Morphological changes in female reproductive organs in the African monarch butterfly, host to a male-killing Spiroplasma

Malmberg Jenny 1
http://orcid.org/0000-0002-0747-7456 Martin Simon H. 2
Gordon Ian J. 3
http://orcid.org/0000-0003-2237-9325 Sihvonen Pasi 1 4
http://orcid.org/0000-0002-7147-5199 Duplouy Anne 1 5 anne.duplouy@helsinki.fi
1 Organismal and Evolutionary Biology Research Programme, University of Helsinki , Helsinki , Finland
2 Institute of Evolutionary Biology, The University of Edinburg, Ashworth Laboratories , Edinburg , UK
3 Centre of Excellence in Biodiversity and Natural Resource Management, Huye Campus , Huye , Rwanda
4 Finnish Museum of Natural History ‘Luomus’, University of Helsinki , Helsinki , Finland
5 Research Center for Ecological Change, University of Helsinki , Helsinki , Finland
Elkelish Amr
Electronic publication date: 2023 Aug 15
Publication date: 2023
Volume: 11
Electronic Location ID: e15853
Received 2023 Mar 17; Accepted 2023 Jul 16
Copyright: © 2023 Malmberg et al.
Copyright year: 2023
Copyright holder: Malmberg et al.
License: This is an open access article distributed under the terms of the Creative Commons Attribution License, which permits unrestricted use, distribution, reproduction and adaptation in any medium and for any purpose provided that it is properly attributed. For attribution, the original author(s), title, publication source (PeerJ) and either DOI or URL of the article must be cited.
License URL: https://creativecommons.org/licenses/by/4.0/

Keywords: Corpus bursa, Signum, Nuptial gift, Sex-ratio distortion, Bacterial symbiosis

Funding: Academy of Finland to Anne Duplouy #321543 Pasi Sihvonen #331995 National Geographic Society to Ian J. Gordon WW-138R-17 Royal Society to Simon H. Martin URF\R1\180682 The project was funded by the Academy of Finland to Anne Duplouy (Grant #321543) and Pasi Sihvonen (Grant #331995), the National Geographic Society to Ian J. Gordon (Grant WW-138R-17) and the Royal Society to Simon H. Martin (Grant URF\R1\180682). The funders had no role in study design, data collection and analysis, decision to publish, or preparation of the manuscript.

==============================
Background

Sexual selection and conflicts within and between sexes promote morphological diversity of reproductive traits within species. Variation in the morphology of diagnostic reproductive characters within species offer an excellent opportunity to study these evolutionary processes as drivers of species diversification. The African monarch, Danaus chrysippus (Linnaeus, 1758), is widespread across Africa. The species is polytypic, with the respective geographical ranges of the four colour morphs only overlapping in East Africa. Furthermore, some of the populations host an endosymbiotic bacterium, Spiroplasma, which induces son-killing and distorts the local host population sex-ratio, creating sexual conflicts between the females seeking to optimize their fecundity and the limited mating capacity of the rare males.

Methods

We dissected females from Kenya, Rwanda and South Africa, where Spiroplasma vary in presence and prevalence (high, variable and absent, respectively), and conducted microscopy imaging of their reproductive organs. We then characterized the effect of population, female body size, and female mating status, on the size and shape of different genitalia characters of the D. chrysippus female butterflies.

Results

We showed that although the general morphology of the organs is conserved in D. chrysippus, female genitalia vary in size and shape between and within populations. The virgin females have smaller organs, while the same organs were expanded in mated females. Females from highly female-biased populations, where the male-killing Spiroplasma is prevalent, also have a larger area of their corpus bursae covered with signa structures. However, this pattern occurs because a larger proportion of the females remains virgin in the female-biased populations rather than because of male depletion due to the symbiont, as males from sex-ratio distorted populations did not produce significantly smaller nutritious spermatophores.

Introduction

Reproductive organs, or genitalia, are under strong selection, which generally leads to considerable variation between species, but relative conservation of the traits within species (House et al., 2013; Langerhans, Anderson & Heinen-Kay, 2016). Consequently, many of these traits can offer diagnostic morphological characters that are of utmost importance for the systematic classification of fauna, but that may still present some variability due, for example, to demographic and/or environmental stochasticity, plasticity, or diverse evolutionary processes, including sexual selection. For example, variations in body size often result in changes in the size of external structures and internal organs in insects (Polilov & Makarova, 2017). Furthermore, although large specimens often hold large organs, the proportion of the changes might vary between organs and the species considered (Polilov & Makarova, 2017). Similarly, sexual selection, either mate-choice, same-sex competition (i.e., sperm competition) and/or male-female conflict (i.e., divergent interests over fertilization), can induce changes in the morphology of genitals within species (Brennan & Prum, 2015; Cordero, 2005; Hosken, Garner & Ward, 2001). For example, in the taurus scarab, Onthophagus taurus (Schreber, 1759), morphological variation of the endophallus sclerites was found to influence male success in populations where females mate multiple times (House & Simmons, 2003; Simmons et al., 2009); while in gerrid water striders, male grasping structures have evolved as a response to male-male competition for female guarding until fertilization of the eggs (Arnqvist & Rowe, 2002).

In Lepidoptera, and in some other insects, males produce spermatophores or ‘mating gifts’, which contain sperm and nutrients that are transferred to the female ovipore during copulation. Each spermatophore can make up to 13% of a male’s body weight (Galicia, Sanchez & Cordero, 2008), and their production by the male is costly and limited. The size of the spermatophore thus typically depends on resources acquired and depleted across the male lifespan (Duplouy & Hanski, 2015; Duplouy et al., 2018; Kaitala & Wiklund, 1994). In contrast, the reproductive organs of female butterflies support the production and fertilization of eggs, for which the reception and digestion of the spermatophore is essential. These organs include the corpus bursae (or bursa copulatrix), a bag-like receptacle for the spermatophore during mating, which inner wall surface is partially covered with sclerotized signa. The function of the signa might vary between Lepidoptera species (Galicia, Sanchez & Cordero, 2008; Xochipiltecatl, Cordero & Baixeras, 2022), but they have often been described as structures involved in the digestion and disruption of the spermatophore envelope (Cordero, 2005; Galicia, Sanchez & Cordero, 2008; Sánchez & Cordero, 2014; Xochipiltecatl, Cordero & Baixeras, 2022). Consequently, the female corpus bursae is likely the stage for intra- and inter-sexual conflicts. Indeed, although large spermatophores would support female fertility, and possibly also prevent remating of the female, they are also costly to produce. In contrast small spermatophores might be distributed between several females, but are unlikely to be sufficient to fertilize all available eggs and provide enough nutrients to stop each female from seeking additional mates and thus allow for sperm competition.

Maternally inherited bacterial symbionts, such as the bacterial taxa Wolbachia and Spiroplasma, are widespread in insects (Ferrari & Vavre, 2011; Vancaester & Blaxter, 2023). They owe their success to their ability to modify their host reproductive system towards their own successful transmission through the host generations. One of the phenotypes these bacteria can induce in their hosts is the selective death of the male offspring at early developmental stages. The so-called male-killing (MK) symbionts have been reported in diverse insect species, including Diptera, Coleoptera and Lepidoptera (Dyson, Kamath & Hurst, 2002; Graham & Wilson, 2012; Hurst et al., 2000; v d Schulenburg et al., 2002), and their prevalence range from 5% to over 95% across the host populations (Charlat et al., 2009; Duplouy et al., 2010; Gordon, Ireri & Smith, 2014). For instance, the blue moon butterfly, Hypolimnas bolina (Linneaus, 1758), and the African monarch butterfly, Danaus chrysippus (Linnaeus, 1758), can host a MK Wolbachia or Spiroplasma (Charlat et al., 2006; Dyson, Kamath & Hurst, 2002; Jiggins et al., 2000).

In infected insect populations, the death of the sons of symbiont-infected mothers often leads to sex-ratio distortions (Charlat et al., 2005; Jiggins et al., 2000; Jiggins, Hurst & Majerus, 2000) that can shape the ecology and evolution of the host species (Engelstädter, Hammerstein & Hurst, 2007; Engelstädter, Montenegro & Hurst, 2004). In addition to possible effects on the host population size and/or on the risk of population extinction (Hurst & Jiggins, 2000), sex-ratio distortions due to MK infections can cause sexual conflicts and shape the evolution of traits involved in these conflicts. For example, because of a locally prevalent MK Wolbachia, the rare H. bolina males from the highly female-biased populations were described as resource depleted because they were in high mating demands (Charlat et al., 2007). In the same populations, a high proportion of the females were also found unmated, while the mated females laid fewer fertile eggs than females from populations where the MK was absent (Charlat et al., 2007). This reduced female fertility in female-biased populations was suggested to have appeared as the females received a limited amount of sperm during copulation (Charlat et al., 2007). In this context, intrasexual conflicts may arise as females race to optimize their fecundity for example via evolving organs that best acquire, accommodate, digest, and convert the spermatophores into resource.

To unravel whether variation in the female reproductive organs is correlated with female size, population, mating status or local prevalence of a MK symbiont, we dissected specimens from several populations across the African range of the butterfly D. chrysippus, including populations where its MK Spiroplasma symbiont was absent or present at high or low prevalence. We showed variations in several characters of the genitalia of mated vs. virgin females, and between populations despite no difference observed in the size of the spermatophores produced by local males. We thus suggest that variation between populations and individuals observed in the size and shape of the female reproductive organs are linked to the plastic response of the corpus bursae to mating opportunity, rather than selective pressures associated with changes in the condition of the local males because of the presence or absence of a MK symbiont in D. chrysippus.

Material and Methods

Samples

The African monarch butterfly, Danaus chrysippus (Lepidoptera: Nymphalidae), is a species belonging to the sub-family Danainae. It is one of the most common and widely distributed butterflies in Africa and Australasia (Hassan, Idris & Majerus, 2012; Idris & Hassan, 2012) and is increasingly common in southern Europe, especially during summer (Koren et al., 2019; Liu et al., 2022). The species is found in many different habitats, such as mountains and deserts, but primarily occurs in open landscapes such as around farmlands (Hassan, Idris & Majerus, 2012). Caterpillars of D. chrysippus often feed on Asclepiadoideae plants, particularly on toxic milkweeds (Asclepias) (Robinson et al., 2010). Four subspecies of D. chrysippus live in separate areas of the overall African species range, but interbreed in one common central hybrid zone (Herren et al., 2007; Liu et al., 2022). In this hybrid region, the MK-symbiont is prevalent and the observed males are rare migrants from uninfected populations surrounding the hybrid zone (Martin et al., 2020; Ndatimana et al., 2022; Smith et al., 2016, 2019).

In this study, we included 67 female specimens from the hybrid zone collected in Rwanda (N = 29) and from two localities in Kenya (N = 38), and 21 female specimens from uninfected populations from three localities in South Africa (Table 1). We thank Nani Croze, Steve Collins, Jody Garbe, Dylan Smith, and Johan Lawson for assisting with access to private lands for sample collection. The Kenyan populations host a male-killing (MK) Spiroplasma symbiont and show high female-biased sex-ratio (Martin et al., 2020). In Rwanda, the MK-inducing symbiont is less common and the sex-ratio distortion remains low but variable through/across the year(s) in that region (Ndatimana et al., 2022). The populations from South Africa are not known to carry the symbiont, and do not show any patterns of sex-ratio distortion (Hassan, Idris & Majerus, 2012; Jiggins, Hurst & Majerus, 2000).

Table 1 Number of female specimens dissected from each population, either treated with KOH for morphometric measurements of the female genitalia, or untreated to collect their spermatophores.

Population	Spiroplasma prevalence	Sex-ratio	Number of samples (N=)	Dissection	
Genitalia	Spermatophore	
Kenya	High	Female biased (Martin et al., 2020; Smith et al., 2019)	38	28	10	
Rwanda	Low	Variable (Rutagarama et al., 2023)	29	10	19	
South Africa	Null	Unbiased	21	12	9	
Total of all populations:	88	50	38	
Note:

Sample size includes specimens collected in the field both as adults or larvae. ‘NA’, Not applicable.

All but a few specimens were collected as adults in the field and killed shortly after collection. The few specimens collected as larvae in the field, originated from two populations (Rwanda and Kenya) and were killed in the laboratory shortly after emerging from their pupae. All specimens were labelled and stored individually in 90% ethanol in the freezer until further manipulated (Martin et al., 2020). Thorax tissue from some of the specimens has been previously used for population genetics and genomics research on the butterfly host (Martin et al., 2020), but all abdomens were intact and of good quality to support the present research.

Samples were collected under the following permits: NACOSTI/P15/3290/3607; NACOSTI/P15/2403/3602 (National Commission for Science and Technology, Nairobi, Kenya); MINEDUC/S&T/459/2017 (Ministry of Education, Kigali, Rwanda); MPB.5667 (Mpumalanga Tourism and Parks Agency, Mbombela, South Africa); FAUNA 0615/202 (Department of Environment and Nature Conservation, Northern Cape Province, South Africa).

Dissections

We prepared the genitalia of 50 specimens according to standard methods used in Lepidoptera (Hardwick, 1950; Robinson, 1976). In brief: the abdomens were heat treated at 94 °C for about 10 min in 10% potassium hydroxide (KOH) to remove fat and other soft tissue. The remaining tissues were cleaned in sterile water under the microscope with the help of small brushes. The abdomens were then individually and carefully cut open laterally with small scissors and tweezers, starting from the base of the abdomen until the seventh abdominal segment. We then detached the genitalia from the basal part of the abdomen by cutting and pulling apart the seventh and eight abdominal segments. The genitalia were then cleaned, coloured with Chlorazol Black, and transferred to 99% ethanol to harden the remaining structures. The female organs targeted in this study were the corpus bursae and the signa. The corpus bursae is a bag-type organ that receives the male’s contribution to reproduction during copulation, the so-called spermatophore or nuptial gift, and the two signa are plates of sclerotised spikes on the internal side of the corpus bursae wall. All dissections were prepared at the Finnish Museum of Natural History in Helsinki (LUOMUS).

The abdomens of the remaining 38 samples, not treated with heat and KOH solution, were opened dry with the help of sterile toothpicks under the microscope to extract the undissolved spermatophores inside of the corpus bursae.

Imaging

The wings of all specimens were imaged using an Epson 10000XL flatbed scanner including a measuring ruler. We photographed each corpus bursae under four different angles (ventral, dorsal, left, and right sides) using the Leica LAS EZ software (version 3.4.0) with the same microscope magnification for each sample. We used the Fixator method as described in Wanke et al. (2019) to fix the female genitalia in the desired position in a petri dish using a nylon thread. The female genitalia and the spermatophores were individually photographed on the side of a graduated scale (millimetre paper or ruler).

Measurements of the wings, the genitalia and the spermatophores

All measurements were done using the software ImageJ version 1.53e (Collins, 2007; Schneider, Rasband & Eliceiri, 2012). All images included a small piece of millimetre paper for scale.

To compare the size of each signum structure between individuals, or against the size of the corpus bursae, or that of the wings, we measured the total area of the corpus bursae (including the area of the appendix bursae, a pouch arising from the tip of the corpus bursae), and the area of each signum (Figs. 1A and 1B). To measure variation in the males’ contribution to mating between populations, we measured the area of each spermatophore dissected out from the corpus bursae of the females (Fig. 1C). To test whether the size of the females affected the size of their reproductive organs, we measured the length between forewing veins CuA1 and CuA2 (Fig. 1D) as a proxi for the size of the butterflies. We chose to measure the length between CuA1 and CuA2 instead of full wing length as the wings were arbitrarily cut off from the thorax for the purpose of other studies (Liu et al., 2022; Ndatimana et al., 2022).

Figure 1 Four types of measurements.

(A) The surface area of the corpus bursae including the appendix bursae, (B) the surface area of the signum, (C) the surface area of spermatophore, and (D) the absolute vein length between forewing veins CuA1 and CuA2 as a proxi for wing size.

Mating status

Several specimens from two populations (seven from Rwanda and two from Kenya) were collected as larvae in the field and remained virgin before being killed in the laboratory. We characterized the shape of the genitalia of those known virgin females, which would have not been distorted by the reception of any male’s spermatophore. The organs from the lab-reared specimens thus provided a visual reference for virgin individuals during the dissection of the field collected specimens, and allowed us to identify whether the adult field-caught females were mated or not. Unfortunately, the KOH treatment dissolved the spermatophore structures within the treated bursae. Thus, we could not directly test how many spermatophores were acquired by the mated females dissected with this method. Additionally, the dissection of the corpus bursae necessary to remove the stored spermatophore(s) led to the destruction of the corpus bursae which size could therefore not be measured in a comparable way. Consequently, female genitalia measurements and spermatophore data came from different individuals.

Molecular work

All molecular work was done at the Molecular Ecology and Systematics lab at the University of Helsinki. We extracted the DNA from one abdominal section using a Qiagen DNeasy Blood & Tissue Extraction Kit (Cat. #69506; Qiagen, Germantown, MD, USA) for 67 specimens individually, while all other samples were screened for the infection for the purpose of an earlier study (Martin et al., 2020). The quality of the DNA extracts was tested by PCR through the amplification of the 5′-end region (~654 bp) of the cytochrome oxidase I (COI) mitochondrial gene using the primers LCO-1490/HCO-2198 (Folmer et al., 1994). To screen for Spiroplasma, we amplified the GDP Spiroplasma gene, using the primer pair GDP1-F/GDPI-R (Martin et al., 2020). Each PCR included a negative control (water) and a respective positive control (Deng et al., 2021).

Statistical analyses

All statistical analyses were performed in R version 4.1.3 (RCoreTeam, 2020). The response variables were individually checked for normality, and log-transformed prior analysis when appropriate (i.e., for largest spermatophore size, or signum area). First, we analysed variation in the distance between forewing veins CuA1 and CuA2 (wing size) using ANOVA with population as a fixed factor.

Then, we analyzed signum area, and corpus bursae area using ANCOVAs with wing size, population, and female assigned mating status (mated vs. virgin), and the interaction between population and mating status, as fixed factors. In order to further assess the specific effects of each population on the female organs size, we corrected the signum and corpus bursae sizes for variation in female size by using residuals from respective linear regressions, and used Tukey’s honest significance tests after using ANOVAs on the wing size-corrected signum area, and wing size-corrected corpus bursae area, with only population and female assigned mating status (mated vs. virgin) as fixed factors, and their interactions. The resulting p-values from the Tukey tests, which must be interpreted cautiously (Darlington & Smulders, 2001; Freckleton, 2002; García-Berthou, 2001), were corrected for multiple testing using a Bonferroni adjustment (α = 0.025).

Finally, we analysed variation in spermatophore area using ANOVAs with only population and female assigned mating status (mated vs. virgin) as fixed factors, and their interactions; and used a Kruskal-Wallis test to analyse the effect of population on the number of spermatophores found per mated female.

Data availability statement

All ecological data and images are available from Zenodo.org: DOI 10.5281/zenodo.7743561.

Results

Wing size

The distance between forewing veins CuA1 and CuA2 (Fig. 1) ranged from 3.504 mm up to 5.867 mm in length. Wing length did not significantly vary between populations (p = 0.88), nor between females of different infection status (p = 0.47, Fig. 2).

Figure 2 Female wing size (measured as the distance between forewing veins CuA1 and CuA2) at each population, and between specimens infected (dashed-lines) or not (full-lines) by a male-killing Spiroplasma.

The boxes represent the interquartile range of the data, and the heavy horizontal lines represent median values, whiskers give the 95% lower and upper percentiles.

Infection status

In total, 37 specimens were found infected with Spiroplasma, including seven from Rwanda (Spiroplasma prevalence = 24%) and 30 from Kenya (prevalence = 79%). All specimens from South Africa were uninfected.

Spermatophores

The spermatophores extracted from the corpus bursae varied in their colours, ranging from orange to silky white; in their shapes, from spheres to flat shapeless shreds (Fig. 3A); and in their size/surface area, from 0.956 to 13.673 mm2 (Fig. 3B). However, the size of the largest spermatophore per mated female (i.e., likely the most recently transferred spermatophore) did not significantly differ between populations (p = 0.18, Fig. 3B).

Figure 3 Spermatophores of various colours, shapes and sizes.

(A) Three spermatophores under the microscope. (B) Surface area (mm2) of the largest spermatophore dissected from each mated female in each population, and (C) spermatophore count per female in each population (data includes mated and virgin females). The boxes represent the interquartile range of the data, and the heavy horizontal lines represent median values, whiskers give the 95% lower and upper percentiles. Coloured circles show the small sample repartition for each population.

The number of virgin females caught as adult in the wild was the highest in Kenya (56%), followed by the Rwanda population (10%), while all females from South Africa were mated (Fig. 3C). The mated females had acquired between 1 and 6 spermatophores, but the spermatophore count per female was not significantly different between populations (Kruskal-Wallis H = 3.605, df = 2, p = 0.17; Fig. 3C, Table 2). On average, we found 1.95 spermatophores per dissected mated female, or 1.54 spermatophores per dissected female (including all field collected adult females, Table 1).

Table 2 Number of mated and virgin females from each population, with the average number of spermatophore per mated female, and average spermatophore count per population, excluding specimens collected as larvae.

Population	Life stage at collection	Mated females	Virgin females	Dissected females	Spermatophore count	
N=	Mean ( x¯) and median ( x~) per mated female only	Population mean ( x¯) and median ( x~)	
Kenya	Adult	4	5	9	11	x¯ = 2.75, x~ = 2 (range: 1 to 6)	x¯ = 1.22, x~ = 0.5	
Larva	NA	1	1	NA	NA	NA	
Rwanda	Adult	9	1	10	17	x¯ = 1.89, x~ = 2 (range: 1 to 4)	x¯ = 1.7, x~ = 1.5	
Larva	NA	7	7	NA	NA	NA	
South Africa	Adult	9	0	9	15	x¯ = 1.67, x~ = 1 (range:1 to 3)	x¯ = 1.67, x~ = 1	
Total	22	6 (+8)	28 (+8)	43	x¯ = 1.95, x~ = 1.5	x¯ = 1.59, x~ = 1	
Note:

‘NA’, Not applicable. Numbers in parenthesis indicate the number of field-collected larvae reared to adulthood in the lab. One dissected specimen from Rwanda was later excluded from the total number of dissected specimens due to hard unidentified material in the female tissues.

Female reproductive organs

We showed that the corpus bursae of D. chrysippus female butterflies was topped by an appendix bursae, and had two signa, one on the ventral side and the second on the dorsal side. The signa were darker than the rest of the corpus bursae due to rows of sclerotized spike-like structures across their surfaces (Fig. 4). The shape and size of the corpus bursae, the appendix bursae and the signa visually varied between our specimens.

Figure 4 The reproductive organs of female Danaus chrysippus butterfly, with the corpus bursae and signa (A and B), a closer-up of the sclerotized spike-like structures of the signa (C), and appendix bursae (D).

The ventral view (A) is a composite of two pictures, resulting from the removal of ventral sclerites, which were obstructing the underlying structures. The lateral view (B) has the ventral sclerites still in place. See Fig. 5 for variation of structures. Pictures (C) and (D) are not in scale relative to pictures (A) and (B).

Figure 5 Reproductive organs of a virgin specimen (from Kenya), and three possibly mated specimens (from Kenya, Rwanda and South Africa), from left to right.

Virgin females showed a folded corpus bursae (A) and a very wrinkly appendix bursae (B), while we suggest that mated females show expanded corpus bursae (C) and appendix bursae (D) filled with spermatophore structure(s) (not visible on the images here). Sample IDs: JM stands for the initials of the first author’s name, followed by a unique sample number.

There was no size difference between the ventral and dorsal signa of a specimen (Fig. S1). Thus, for simplicity, we only used the values from the ventral signum for the following analyses. Additionally, although larger females showed slightly larger corpus bursae, the difference was not statistically significant (p = 0.15); however, larger females showed significantly larger signa (ANOVA, df = 1, F-value = 4.7, p = 0.037, Fig. S2).

Population comparison

Population had a significant effect on the size of the signum (ANOVA, df = 2, F-value = 5.5, p < 0.01), the size of the corpus bursae (ANOVA, df = 2, F-value = 22.7, p < 0.01), and on the ratio between the size of the signum and the corpus bursae (ANOVA, df = 2, F-value = 5.5, p < 0.01). Furthermore, females from Rwanda showed the largest corrected corpus bursae (Rwanda vs. Kenya, Tukey test, p < 0.01, 95% CI [3.69–7.99]; Rwanda vs. South Africa, Tukey test, p = 0.014, 95% CI [−6.03 to −0.97]); while the corpus bursae (corrected) from females from South Africa and Kenya were similar (p = 0.078; Fig. S1A). Females from Rwanda also showed larger signa (corrected) than females from Kenya (Tukey test, p < 0.01, 95% CI [0.28–1.64], Fig. S1B), all other population comparisons were not significantly different (p > 0.17).

A small ratio between the signum area and the corpus bursae area would mean that the signa cover a small surface of the bursa, while a large ratio would mean that the signa cover a large surface of the bursa. The ratio between signum area and corpus bursae area was also significantly different between populations (ANOVA, df = 2, F-value = 16.6, p < 0.01). The ratio between signum area and corpus bursae area was significantly smaller in females from Rwanda compared to the females from Kenya (Tukey test, p < 0.01, 95% CI [−0.16 to −0.06]), but not when compared to the females from South Africa (p = 0.12, Fig. 5). The ratio between signum area and corpus bursae area was also significantly smaller in the females from South Africa compared to the females from Kenya (Tukey test, p < 0.01, 95% CI [−0.11 to −0.02], Fig. 5).

Virgin vs. mated females

We had specimens collected as larvae from two populations (Rwanda and Kenya), which provided adult virgin specimens. The corpus bursae and associated signa and appendix bursae from these specimens were compact, folded, and highly wrinkled (Fig. 6). Based on these observations, we suggested that field collected specimens with expended corpus bursae of orange colour, and signa of beaver-tail shape, were mated females (Fig. 6); while others showing clear, compact, folded, and wrinkled organs were virgin. The number of estimated virgin females was large in the Kenyan population (Nvirgin = 14, Nmated = 12).

Figure 6 Variations in the ratio between signum and corpus bursae area between virgin and mated females from three populations across Danaus chrysippus natural range.

The boxes represent the interquartile range of the data, and the heavy horizontal lines represent median values, whiskers give the 95% lower and upper percentiles.

Using only field collected adult females, we showed that the surface area of the corpus bursae varied between 2.65 and 8.0 mm2 for virgin females, and between 5.6 and 15.2 mm2 for mated females, while the surface area of the signa varied between 1.15 and 3.4 mm2 for virgin females, and between 1.98 and 4.86 mm2 for mated females (Fig. S1). The surface area of the corpus bursae was significantly different between virgin and mated females (ANOVA, df = 1, F-value = 29.6, p > 0.01). It was significantly different between virgin and mated females from Kenya (Tukey test, p < 0.01, 95% CI [0.99–6.36]), and from South Africa (Tukey test, p < 0.01, 95% CI [1.03–10.33]). The surface area of the signa was also significantly different between virgin and mated females (ANOVA, df = 1, F-value = 19.7, p > 0.01). In particular, it was significantly different between virgin and mated females from South Africa (Tukey test, p < 0.01, 95% CI [0.29–2.87]), but not from Kenya after correction for multiple testing (p > 0.025). There was no measure for virgin females from Rwanda. The ratio between the signum area and the corpus bursae area was larger in the virgin females (ANOVA, df = 1, F-value = 18.9, p < 0.01, Fig. 6).

Finally, there was no significant interaction between population and mating status of the females on neither of the three traits (p > 0.11).

Discussion

According to our knowledge, we provided the first images and study of the female reproductive organs of the African Monarch D. chrysippus. Consistent with schematic drawings by Mal et al. (2015), female D. chrysippus butterflies have two signa. Each signum is of similar size, of a beaver-tail shape, and covers on average 32% of each side of the corpus bursae, but does not extend into the appendix bursa.

We demonstrated that in D. chrysippus the size of the corpus bursae, that of the signa, and their ratio, varied with the mating status of the females regardless of their population of origin. In each population, the virgin females showed smaller organs, while mated females showed expanded organs. Comparative illustrations of virgin vs. mated female genitalia are scarce in insects and other arthropods (Mouginot et al., 2015; Sihvonen & Mikkola, 2002), but there is evidence for the female organs to vary in their shape, size and possible functionality after mating. For example, in seed beetles (Coleoptera: Bruchidae), male genitalia are armed with sclerotized spikes that serve as anchors to the female during copulation that cause scar-tissues to be observed in mated females only (Crudgington & Siva-Jothy, 2000; Edvardsson & Tregenza, 2005). Similarly, in the orb-weaving spider, Larinia jeskovi (Marusik, 1987), the male removes a coupling device (i.e., scapus) from the female external genitalia after copulation, inhibiting the possibility for the female to remate (Mouginot et al., 2015). In D. chrysippus, we suggest that mated females showed larger organs because they were filled up with one to several spermatophores from their mate(s). Unfortunately, the KOH treatment of the female organs destroyed the spermatophores within the bursae in D. chrysippus. This has challenged our ability to obtain both the data on males’ contribution and female genitalia traits from the same individuals as it does not allow (I) to determine with certainty which females were mated or not before dissection, although lab-reared virgin individuals were informative, (II) to determine how many times each female had mated before dissection, and (III) to evaluate the size and composition of the male’s contribution to mating in each population.

The signum structures coupled with muscles associated with the corpus bursae (Allman, 1930) have been described as ‘lamina dentata’ (Petersen, 1904), a structure possibly involved in the digestion, by grating or perforating of the nutrient-rich spermatophores after copulations (Cordero, 2005; Galicia, Sanchez & Cordero, 2008; Xochipiltecatl, Baixeras & Cordero, 2021). If this is true in D. chrysippus, we expected that natural selection will act on the female genitalia in response to sexual conflict. In polyandrous butterfly species, such as D. chrysippus, we expect males to transfer large spermatophores that can act as mating plugs in the receiving females (McNamara, Elgar & Jones, 2009; Wedell, 1993). In response, females might evolve organs that efficiently digest each nuptial gift to allow for multiple mating and the avoidance of the fertilization of all the eggs by a unique genitor (Lewis et al., 2020). Additionally, as the nutrients received from the males can be upcycled towards the production of eggs, or of better-quality eggs (Wedell & Karlsson, 2003), females might engage in female-female conflicts, and evolve organs that also optimise the intake from the nuptial gift (Meslin et al., 2017), especially in population where males transfer small spermatophores.

We found that the surface area of the profile view of the female corpus bursae, that of the signa, and their ratio, were different between populations of D. chrysippus. However, evidence show that this variation is unlikely due to variation in the prevalence of the MK Spiroplasma between populations. First, in the Kenyan population the proportion of mated females showing expanded reproductive organs is low compared to the other populations, in accordance with another recent study (Rutagarama et al., 2023). Although this is concordant with previous studies showing that, after a certain threshold, the shortage of males can increase the risk of local females remaining virgin (Charlat et al., 2007), the percentage of males was not shown to corelate with the percentage of virgin females in the Kenyan population (Rutagarama et al., 2023). Additionally, although we showed that the spermatophores vary in size between D. chrysippus specimens, the size variation was not significant between populations with different prevalence levels of the MK Spiroplasma symbiont. However, we acknowledge our sample size might have been too small to properly test the link between the size of the spermatophores and that of the female reproductive organs. Although resource depletion in males after several matings has been documented in diverse Lepidoptera species (de Morais, Redaelli & Sant’Ana, 2012; Duplouy et al., 2018; Sims, 1979); the study in the blue-moon butterfly by Charlat et al. (2007) remains the only known example of populations wide resource depletion in males due to the prevalence of a MK symbiont. Clearly, additional studies in other Lepidoptera are needed to test whether male resource depletion is common in species infected with MK symbionts, and whether changes in spermatophore size between populations can indeed drive local changes in the female reproductive organs.

The location of the genes responsible for the female reproductive characters will likely influence the possibility that changes in those traits might be driven by selection (Charlesworth, Coyne & Barton, 1987; Rice, 1984). In Spiroplasma-infected D. chrysippus, the W female chromosome is fused to an autosome, and is called the neo-W chromosome (Smith et al., 2016; Smith et al., 2019). If the genes coding for the signa and/or bursae were to be located on the neo-W, there could be a strong role for selection in the low sex ratio Kenyan populations, since this chromosome is matrilineally inherited and is found in up to 95% of all females (Martin et al., 2020). However, if these genes were to be located on any other chromosome, the dilution effect of incoming genes (50%) from immigrant males that are likely to have come from high sex ratio populations, together with their subsequent elimination with dead males in each generation, may considerably impede such selection.

Morphological traits such as the count and shape of the signa and the smoothness or complexity of their surface can provide taxon-specific diagnostic characters as they vary enormously among Lepidoptera species (Scoble, 1995). The signa have for example been described as smooth, or ornamented with micro-protuberances of different ornamented shapes (i.e., spikes, teeth, spines, horns, bands, patches, or plates, Galicia, Sanchez & Cordero, 2008). We showed that both signa are covered with spike-like sclerotized structures of similar size, which give the signa their darker colour compared to the rest of the bursa. In comparison, the closely related Monarch butterfly, D. plexippus (Linnaeus, 1758), has a large, pear-shaped corpus bursae with two large signa, each covered with bands of heavily chitinized micro-protuberances pointed in opposite directions from the median (Rogers & Wells, 1984; Urquhart, 1960), while Mal et al. (2015) only described the bursae in the striped tiger butterfly, D. genutia (Cramer, 1779), as a large balloon-shaped bag with two rod-life sclerotized signa. The female organs thus seem similar between these three Danaus butterfly species; however, in general the bi-signate condition is uncommon in Lepidoptera. For example, in geometrid moths, the two signa character is rare (Murillo-Ramos et al., 2021 and references therein). However, the lack of extensive morphological revision describing the female genitalia of Nymphalidae or Danainae butterflies, where D. chrysippus is classified, does not currently allow the use of these morphological results in a wider evolutionary framework for these butterflies, contrasting for instance with the work done in Tortricidae (Lincango, Fernández & Baixeras, 2013). Furthermore, although it has been suggested from few other Lepidoptera species (Xochipiltecatl, Baixeras & Cordero, 2021), whether and how the digestion of the spermatophore occurs, and whether the signa and bursae are indeed involved in the mechanical digestion of the spermatophore in D. chrysippus remains to be fully experimentally tested.

Supplemental Information

Supplemental Information 1 Variations in the corpus bursae and signum area between virgin versus mated females from three populations across Danaus chrysippus natural range.

The boxes represent the interquartile range of the data, and the heavy horizontal lines represent median values.

Click here for additional data file.

Supplemental Information 2 Signum area size increases with female wing size (p=0.037).

Click here for additional data file.

Thanks to Daisuke Kageyama for early discussions around the project. The first author would like to thank Ida-Maria, Torgny, David, Viivi and Kati for support and discussions on the project. The molecular research was performed at the MES lab facilities, dissections and imaging were performed at the Finnish Museum of Natural History (LUOMUS) at the University of Helsinki, Finland. We thank Reinier Terblanche and David Smith for contributing to sample collection; Reinier Terblanche and Jeremy Dobson for assisting with permit acquisition; Niklas Wahlberg, the members of the Life-History Evolution and Insect Symbiosis research groups at the University of Helsinki, Dr. Carlos Cordero and anonymous reviewers for discussion on the manuscript. Portions of this text were previously published as part of a preprint (Malmberg et al. 2022).

Additional Information and Declarations

Competing Interests

Author Contributions

Field Study Permissions

Data Availability

The authors declare that they have no competing interests.

Jenny Malmberg performed the experiments, analyzed the data, prepared figures and/or tables, authored or reviewed drafts of the article, and approved the final draft.

Simon H. Martin conceived and designed the experiments, authored or reviewed drafts of the article, collected samples, and approved the final draft.

Ian J. Gordon conceived and designed the experiments, authored or reviewed drafts of the article, collected samples, and approved the final draft.

Pasi Sihvonen conceived and designed the experiments, performed the experiments, prepared figures and/or tables, authored or reviewed drafts of the article, and approved the final draft.

Anne Duplouy conceived and designed the experiments, analyzed the data, prepared figures and/or tables, authored or reviewed drafts of the article, and approved the final draft.

The following information was supplied relating to field study approvals (i.e., approving body and any reference numbers):

Samples were collected under the following permits: NACOSTI/P15/3290/3607; NACOSTI/P15/2403/3602 (National Commission for Science and Technology, Kenya); MINEDUC/S&T/459/2017 (Ministry of Education, Rwanda); MPB.5667 (Mpumalanga Tourism and Parks Agency, South Africa); FAUNA 0615/202 (Department of Environment and Nature Conservation, Northern Cape Province, South Africa).

The following information was supplied regarding data availability:

Raw data available on Zenodo:

Malmberg, Jenny, Martin, Simon H., Gordon, Ian J., Sihvonen, Pasi, & Duplouy, Anne. (2022). Infection with a male-killing Spiroplasma bacterium might drive morphological changes in female reproductive organs in a butterfly. Zenodo. https://doi.org/10.5281/zenodo.7743561.

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
