# Peer review of "Morphological changes in female reproductive organs in the African monarch butterfly, host to a male-killing Spiroplasma"

_PeerJ, doi:10.7717/peerj.15853_

## Round 0.1 · original submission · Major Revisions

We can accept your paper after a major revision.

·

Excellent Review

This review has been rated excellent by staff (in the top 15% of reviews)
EDITOR COMMENT
positive, clear and accurate

Basic reporting

The subject of this manuscript is fascinating and the data is interesting. Unfortunately, in my opinion, the limited sample size and, specially, the nature of the data pose serious limitations on the interpretation of the results. I think that he hypothesis advanced by the authors is not supported by the trends in their own data (see Validity of the findings). On the other hand, the presentation of the manuscript can be improved because I found a number of inconsistencies and errors.

Here I will refer only to the main errors and problems in the “structure” of the manuscript. Smaller problems and suggestion are in the sticky notes in the review copy attached.

1. Both in the abstract (line 23) and the introduction (lines 81-83), the authors mention sexual conflict resulting from the effect of the male-killing Spiroplasma on sex ratio without providing a clear explanaition. In both cases, it seems more likely that the effect is on female competition for males (specially for the virgin males that, in several other species, produce the largest spermatophores).

2. In the abstract (line 30), it is mentioned “The small virgin females have the smallest organs, while the same organs were expanded in mated females”. This is not true, the CB was not affected by female size and, to my mind, the spelling of this sentence suggests that virgin females are smaller tan mated females, which is not true.

3. I think that it would be better for the understanding of the manuscript if the genital structures studied (S, CB and appendix bursae) and their functions (if known) were explained in the last paragraph of the introduction. In the manuscript they are mentioned in separate parts (CB in the context of the dissection methods in line 138, and S and appendix bursae in the Measurements section) and the structure and possible function of the S is not mentioned before the Results.

4. In the Results, figure 3 is mentioned for the first time before the first mention of figure 2.

5. In line 226 it is mentioned that “the corpus bursae…was often topped by an appendix bursae”, indicating variation in the presence of the appendix bursae. Although this is not impossible, it would be quite peculiar and interesting. However, no further mention of this is found in the paper. I would expect to know the proportion of females with and without appendix bursae, as well as the distribution of females with (and without) appendix bursae depending on mating status and population.

6. In the Discussion (in line 294) it is mentioned that the signa are involved in the digestion of the “nutrient-rich surface of the spermatophores”. Although there could be some nutrients in the “surface”, the nutrients and other components (for example protective and hormone-like substances) are located in the contents of the spermatophore (in fact, spermatophore envelopes often remain within the CB without being digested).

7. Considering the relatively small sample sizes, I suggest that in Table 2 the authors present median values instead of averages, and I think it is necessary to present some measure of variation (minimum-maximum or interquartile range).

Experimental design

As I mention in the “Validity of the findings” section, the main problem is the use of females of unknown mating history both for the comparative study of their genitalia (somewhat problematic) and for the study of spermatophore size (very problematic).

Validity of the findings

The hypothesis proposed is not supported by the results. I need to summarize the results to explain why. The authors measured the corpus bursae (CB) and signum (S) of 51 females collected in the field in three populations differing in the level of the male-killing Spiroplasma infection and, thus, in sex ratio bias, as well as of eight virgin females obtained from reared field-collected larvae; preparation of these specimens destroyed the contents of the CB and thus it is not known if they contained or not spermatophores. The authors also counted and measured the spermatophores contained in the CB of 35 field-collected females that were not used for measuring genital structures (CB and S). By comparing the CB of virgin females and mated females (those used for spermatophore counting), the authors made a reasonable assignation (reasonable because the differences observed in size and shape were large) of the females used for genital measurements into (potentially) virgin and (potentially) mated. The main results were: (i) CB and S of mated females were larger than those of virgin females, as expected if the CB remains at least partially distended after the digestion of the spermatophores (either naturally by the female or, artificially, by the KOH used for sample preparation). In contrast, (ii) the ratio [S area/CB area] was larger in virgin females. (iii) Across populations, as expected, the percentage of virgin females was directly proportional to the prevalence of Spiroplasma (and, thus, inversely proportional to the percentage of males in the population). Intriguingly, (iv) across populations, the average number of spermatophores per mated female was directly related to the proportion of virgin females, an result not discussed by the authors (male mate choice or competitive differences between females, or something else, could explain this result). (v) Comparison of the largest spermatophores found in the mated females did not show between population differences in spermatophore size. (Note of the reviewer: the largest spermatophore is usually the last received, when previous spermatophores were at least partially digested; in general, digestion of spermatophores starts a few hour after mating. Thus, although it sounds reasonable to compare the size of only the largest spermatophores, even these could vary due to different degrees of digestion.)

From these results, the authors propose that: “Females from highly female-biased populations, where the male-killing Spiroplasma is prevalent, also have a larger area of their corpus bursae covered with signa structures. These results suggest that male depletion due to the symbiont, might result in smaller spermatophores, and select for female genitalia features that optimize the digestion of small nutritious spermatophores”. However, this suggestion is unfounded because the authors did not find differences in spermatophore size (see v above). Furthermore, the idea that the S cover a larger proportion of the CB in the population with a larger prevalence of Spiroplasma comes from a calculation obtained from measurements made in all the field collected females, both (actual and potentially) virgin and (potentially) mated. The problem with this calculation is that in the Kenya population (highest prevalence of Spiroplasma) more than half of the females were virgin vs. 10% and 0% in Rwanda and South Africa, respectively. Since virgin females have a larger S/CB ratio (see ii above), the larger proportion of virgin females in Kenya could give the (possibly wrong) idea that in Kenya females have a larger S/CB ratio, instead of an adaptation to males producing small spermatophores (for which, as explained before, there is also no evidence in this study).

Additional comments

No comment

Reviewer 2 ·

Excellent Review

This review has been rated excellent by staff (in the top 15% of reviews)
EDITOR COMMENT
clear and straightforward - positive and helpful

Basic reporting

Yes, the manuscript uses clear, professional English. Yes, sufficient context is provided with the minor exception of a few comments listed by line below. Yes the overall structure, tables, figures are appropriate and data are shared. Yes, it is self-contained with results relevant to hypotheses.


Line 44: Can you elaborate or clarify what you mean by “due to stochasticity” here? Developmental stochasticity? Evolution in response to environmental stochasticity?

L48-49: Wouldn’t sperm competition simply be one form a sexual selection?

L87-88: In this line you lay out your purpose but to me it seems as if you have not yet connected the main points from the previous paragraphs to the goals laid out in lines 87-88 as clearly as I think you should. Can you provide a more detailed rationale for why you think that MK symbionts would result in morphological changes in female reproductive organs? I see previously that you mention the connection between the MK symbionts and sexual conflict, and the connection between sexual conflict and a vague idea of possible changes in females, but what changes might you expect to see and why? My reason for requesting this clarity is not because I don’t accept that a valid connection can be made but simply that I think it aids the reader in understanding better the context of what is to follow. To me, having a clear idea of why each trait was measured, as I read the methods and results, allows me to read with better focus.

L138: You write that females organs, plural, were targeted but then you list only the corpus bursa. Can you make this more clear?

Experimental design

This is somewhat exploratory research, so the research questions are not terribly specific. See my earlier comments about lines 87-88. That said, the methodological approach is appropriate for the questions, and with the exception of my questions about the statistical analyses, the methodology is clear. I list a few questions, by line number, below. Most important, in my opinion, are those dealing with how the results were analyzed.

L162: Wouldn’t the size of the spermatophore vary depending upon not just the size provided by the male but also factors like time since deposition (if, for example, it has been substantially drained or digested)?

L165: Is the idea of measuring genitalia of mated and unmated females to determine if the mating act itself causes variation? Or is the idea that females with certain attributes are more likely to be mated than are females with other attributes? And how could you distinguish between these two explanations for differences in genitalia between mated and unmated females?

L191-193: In your analyses of signum area and corpus bursa, it appears you account for female size by including wing size as a covariate (which I would argue is the proper approach). But then in your next lines you refer to “size-corrected” traits (and also in lines 235-236) but it is not clear to me what this means. Does this simply refer to the analyses with the wing size covariate? Or are you using ratios (e.g. trait divided by wing size)? Part of my confusion here is that including the covariate is the proper way to “size correct” but you appear to be describing two different sets of analyses but I cannot tell for sure.

Because I am not sure if you used ratios or not, I will add some comments on the use of ratios to correct for body size, which you might choose to ignore but perhaps they could be helpful.

The use of ratios is a fairly common way to try to control for overall body/wing size differences in butterfly research and so in that sense, it is understandable that the authors might wish to do the same here for the sake of consistency. However, there is also a fairly large literature, mostly from the fields of physiological ecology and systematics, that is critical of the use of ratios (and similar indices) as a means for “controlling for” body size because of undesirable statistical properties that emerge when the trait of interest and overall size do not scale isometrically. An easy place to jump into the literature here would be papers by GC Packard and TJ Boardman, most from the late 1980s or early 1990s. Not all authors agree with Packard and Boardman’s criticisms, but many do so I think the authors of the current manuscript should consider either arguing for their method in light of these potential problems or consider an alternative approach. Typically, the use of overall size as a covariate in an ANCOVA-type of analysis is recommended.

More generally, I think much more clarity in how the statistical analyses were conducted would be helpful. For example, in your analyses did you include wing size, population and mating status as independent variables in that order in the model? If so, is there a particular reason you chose that order? Did you consider analyses with the predictors in a different order? Or, if you chose a method (e.g. Type 3 sums-of-squares instead of the default Type 1 in R) where order of predictors does not matter, can you specify that? Did the models include interactions? Can you explain your rationale for either including or not including interactions? Can you elaborate on how the Bonferroni was applied, i.e. was it across every p-value in the manuscript? If not, what constituted the family of tests to which the adjustment was applied?

Lastly, I prefer to see both treatment and error degrees of freedom reported with the F-tests. In other words, not just df = 2, but df = 2,X, where X is the number of error or residual degrees of freedom. This gives the reader a quick approximation of sample size, if nothing else. (see, e.g. line 206)

Validity of the findings

Outside of the questions raised about how the analyses were conducted, the results appear valid. Generally speaking, I found the discussion to be clearly written. The authors describe what they think their results mean, and they acknowledge in appropriate places some of the limitations of their findings. For example, they acknowledge the possibility of low statistical power, and limits to the ability to infer cause and effect from their data.

Additional comments

Because this work was mostly non-experimental (i.e. did not involve direct experimental manipulation) the ability to infer cause and effect is limited, though as I said above the authors acknowledge this. But that does mean that the paper is to a large degree a description of patterns that could have any number of causes. In addition, the paper addresses questions that are probably useful to a fairly limited number of readers of a general biological journal. In other words, detailed description of butterfly genitalia are very useful for some biologists, but mostly of little interest to non-specialists. The inclusion of the MK symbionts, particularly as they might affect ecological factors like sex ratios and therefore have indirect effects on the reproductive organs is an interesting addition. This might make the content more appealing to a (somewhat) broader audience. That said, I can also imagine the paper might get more attention from readers if it appeared in an entomological journal.

Reviewer 3 ·

Basic reporting

.

Experimental design

.

Validity of the findings

.

Additional comments

In this study the authors explored the effects that an infection of a Spiroplasma bacterium has on the morphology of the female sexual organs in a species of a butterfly. Overall, the study is well done and the manuscript is very well written. The topic is extremely interesting as this is something that is not well known and publishing this study will fill a gap in our knowledge of this phenomenon and will hopefully stimulate more research into this question. There are some small issues that need to be fixed in the manuscript – I made notes on in the pdf itself. The figures are really nice overall!

Annotated reviews are not available for download in order to protect the identity of reviewers who chose to remain anonymous.

---

## Round 0.2 · Minor Revisions

You manuscript will be accepted after minor revision.

·

Basic reporting

I just found three minor issues in the revised version:

--Lines 347-349: The sentence "In response, females might evolve..." needs references.

--L. 366: change "mating" for "matings".

--L. 583-584: This reference is misplaced. It must be moved up, between Rutagarama et al. (2023) and Schneider et al. (2012)

Experimental design

Ok

Validity of the findings

Ok

Additional comments

All my suggestions and criticisms were properly addressed in the revised version.

Reviewer 2 ·

Basic reporting

I have no new substantive comments about the basic reporting.

Experimental design

Lines 217-226: In this section, the authors describe their statistical methods for accounting for overall body size (using the wing size measure as a proxy). In the earlier manuscript, I asked for clarification on how this was done. I do believe that I now understand better what has been done. That said, it still appears that the authors have chosen a common, but not quite appropriate, method. In fact, it looks like they conduct the proper analysis (a model that includes the factors of interest, i.e. population and mating status) and the size covariate first. But then it appears they choose a follow-up analysis wherein they use the residuals from regressions of trait on wing size (“from respective linear regressions”, line 221) as the data in analyses that are almost identical to the previous one, but which now leave out the wing size variable (because the residuals now serve as the “size corrected” response variable).

As I said, this is a very common approach to dealing with body size but as with ratios, it has problems, and these problems are generally avoided by simply interpreting the results appropriately from the model that contains the size covariate.

In short, assuming the authors used a linear model with wing size entered first, and then population and then mating status (as described in lines 217 to 219), then when the significance test for population is conducted, it has already accounted for wing size. When the mating status test is conducted, it tests for any effects after both wing size and population has been accounted for (since mating status is last). In other words, the authors have already controlled for wing size when they assessed population (and mating status) in their model that included wing size. The analysis using the residuals is simply doing something that they already did, but is now doing it less appropriately (see below). In the end, I would not be surprised if the two approaches give similar results but it is possible that they will not. See the list of papers below for the ways in which this might matter. The Darlington and Smulders paper, in particular, shows how the “residual analysis” can be either more or less conservative than the proper “ANCOVA” approach. So the good news here is that first, it’s unlikely that the conclusions will change and second, the proper analysis is not only easy to conduct but the authors have already done it. There’s just no reason to include the additional improper one.

This issue has been explained in fairly non-technical ways in a series of articles published in the evolutionary ecology literature. I would recommend the authors read “On the misuse of residuals in ecology: regression of residuals vs. multiple regression” by Freckleton, from the Journal of Animal Ecology, May 2002 (volume 71, pages 542-545). In that paper, the author also cites two other papers that address the same basic procedure but point out other limitations. These are the papers by Garcia-Berthou (2001) called “On the misuse of residuals in ecology: testing regression residuals vs. the analysis of covariance” (also from the Journal of Animal Ecology, 2001) and that by Darlington and Smulders called “Problems with residual analysis” in Animal Behaviour, 2001, 62:599-602.

Validity of the findings

No comment

Reviewer 3 ·

Basic reporting

I am happy with the revised version and consider it acceptable for publication.

Experimental design

Research in this study is well performed.

Validity of the findings

The authors explain their findings carefully and also discuss the potential shortcomings of their approach.

---

## Round 0.3 · accepted · Accept

We confirmed that the authors have addressed all of the reviewers' comments and the manuscript is ready for publication.